# Unleashing the power of Neural Collapse for Transferability Estimation

## Abstract

Transferability estimation aims to provide heuristics for quantifying how suitable a pre-trained model is for a specific downstream task, without fine-tuning them all. Prior studies have revealed that well-trained models exhibit the phenomenon of Neural Collapse. Based on a widely used neural collapse metric in existing literature, we observe a strong correlation between the neural collapse of pre-trained models and their corresponding fine-tuned models. Inspired by this observation, we propose a novel method termed Fair Collapse (FaCe) for transferability estimation by comprehensively measuring the degree of neural collapse in the pre-trained model. Typically, FaCe comprises two different terms: the variance collapse term, which assesses the class separation and within-class compactness, and the class fairness term, which quantifies the fairness of the pre-trained model towards each class. We investigate FaCe on a variety of pre-trained classification models across different network architectures, source datasets, and training loss functions. Results show that FaCe yields state-of-the-art performance on different tasks including image classification, semantic segmentation, and text classification, which demonstrate the effectiveness and generalization of our method.

## 1 Introduction

Transfer learning has evolved into a mature field in recent years. The "pre-training then fine-tuning" has become a standard training paradigm (Ding et al., 2023) for numerous tasks in the realm of deep learning and diverse repositories of pre-trained models, known as model zoos, are established[1]. These models are constructed through combinations of diverse network architectures, source datasets, and loss functions. Transferability estimation (Bao et al., 2019; Tran et al., 2019) aims to find a metric to indicate how well the pre-trained models perform on a given target dataset without fine-tuning them all. This purpose is non-trivial and task-adaptive, and an effective transferability metric should exhibit a high correlation between the score calculated for each pre-trained model and its performance after fine-tuning.

Classical literature (Papyan et al., 2020) indicates that for a well-trained model, the phenomenon known as Neural Collapse (NC) being more pronounced corresponds to better model performance. Specifically, with high NC levels, features should exhibit the following characteristics: 1) separation between classes; 2) compactness within each class; 3) equiangularity between each pair of class distributions (*i.e.*, distribute at the vertices of a simplex Equiangular Tight Frame). The convergence of models towards NC usually results in the improvement of out-of-sample model performance and robustness to adversarial examples (Papyan et al., 2020). However, this commendable property generally occurs in the well-trained models, *i.e.*,

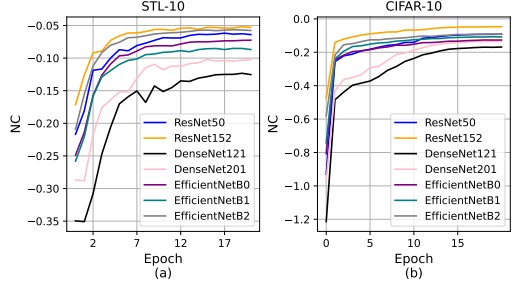

Figure 1: Observation of Neural Collapse during model fine-tuning on (a) STL-10 and (b) CIFAR-10.

---

[1] pytorch.org/hub; docs.openvino.ai/; tfhub.dev/

fine-tuned models, rather than the pre-trained models. We further explore the relationship of NC between the pre-trained models and their corresponding fine-tuned models. To be specific, based on a rough metric of NC (Zhu et al., 2021; Li et al., 2022), we fine-tune several heterogeneous models pre-trained on ImageNet on two different target datasets, and track the changes in their NC scores. As shown in Fig. 1, we find that the NC score ranking in these pre-trained models remains mostly consistent during fine-tuning. This observation inspires us to measure the neural collapse of the pre-trained models for addressing the task of transferability estimation.

Previous works (Papyan et al., 2020; Zhu et al., 2021; Li et al., 2022; Tirer & Bruna, 2022) that study the Neural Collapse phenomenon usually rely on the first two of three characteristics as a measure of NC. This is because the three characteristics of NC typically occur simultaneously in well-trained models. Many existing works in transferability estimation also take into account the first two points (Bao et al., 2019; Pándy et al., 2022; Thakur et al., 2022), and some endeavors also incorporate additional factors such as feature informativeness (Bao et al., 2019). However, the neglect of the last characteristic may be deemed acceptable for well-trained models, but it is not applicable to pre-trained models that have not been fine-tuned on target data. It could potentially result in the selection of models that exhibit biases towards specific classes.

In this paper, we propose a novel transferability estimation metric termed Fair Collapse (FaCe). FaCe consists of two key components: variance collapse term and class fairness term. The variance collapse term is calculated based on the magnitude of between-class covariance compared to within-class covariance. For the class fairness term, we first calculate the overlap between all pairs of class distribution to construct an overlap matrix. Afterward, we apply temperature scaling and a softmax function to this matrix and compute its entropy as our class fairness term. A higher entropy signifies the class distributions exhibit a more even spread in the feature space. This indicates that the model is fair to all classes and does not exhibit biases towards specific classes. Finally, both the variance collapse term and class fairness term are min-max normalized individually to alleviate the impact of different scales and summed to yield the final FaCe score.

Overall, the main contribution can be summarized as follows: 1) We explore the impact of Neural Collapse (NC) in the "pre-training then fine-tuning" paradigm and observe that the ranking of NC in the pre-trained models remains mostly consistent during the fine-tuning process. This observation inspires us to estimate the transferability by measuring the neural collapse of pre-trained models. 2) We introduce a novel metric Fair Collapse (FaCe) to estimate the transferability of pre-trained models. FaCe simultaneously takes into account the cues of separation between classes, compactness within each class, and fairness of the pre-trained model towards each class together. 3) To validate the effectiveness and generality of FaCe, we perform experiments on both computer vision (image recognition, segmentation) and natural language processing (text classification) tasks. We also consider various training paradigms for pre-trained models, including multiple model architectures, multiple loss functions, and multi-source datasets. Experimental results demonstrate that FaCe yields competitive results for transferability estimation.

## 2 RELATED WORKS

**Transferability Estimation.** With the advent of the era of large AI models, the selection of appropriate models for downstream tasks has become a critical issue. Consequently, there has been an increasing amount of research in the field of transferability estimation. The Bayesian-based methods (Nguyen et al., 2020; Tran et al., 2019; Li et al., 2021; Agostinelli et al., 2022) measure the domain gap between the source and target from a probabilistic perspective. Take two typical examples, LEEP (Nguyen et al., 2020) is the classification performance on the Expected Empirical Predictor (EEP); NCE (Tran et al., 2019) considers the conditional entropy between the label assignments of the source and target tasks. Information theory-based methods (You et al., 2021; Bolya et al., 2021; Tan et al., 2021) measure the information contained within features. LogME (You et al., 2021) is the maximum value of label evidence (marginalized likelihood) given extracted features. OTCE (Tan et al., 2021) uses optimal transport to estimate domain difference and the optimal coupling between source and target distributions. TransRate (Huang et al., 2022) measures the transferability as the mutual information between features of target examples extracted by a pre-trained model and their labels. Additionally, feature structure-based methods (Bao et al., 2019; Pándy et al., 2022; Thakur et al., 2022) set different metrics based on the feature space structure of pre-trained models on the

target dataset. H-score (Bao et al., 2019) considers between-class variance and feature redundancy. GBC (Pándy et al., 2022) is the summation of the pairwise class separability using the Bhattacharyya coefficient. Our method is a typical feature structure-based method, and compared to existing methods, we further consider the class fairness of pre-trained models towards target classes. NCTI (Wang et al., 2023) is a concurrent work of our method, which is also inspired by neural collapse. NCTI consists of three terms, which correspond to three characteristics of neural collapse, respectively. Different from NCTI, which assesses the geometry structure based on nuclear norm, our method is based on the class distribution overlapping matrix entropy.

**Neural Collapse (NC).** Existing work (Papyan et al., 2020) exposes a pervasive inductive bias in the terminal phase of training (TPT) called Neural Collapse. TPT begins at the epoch where the training error first vanishes, which is a sign of the completion of model training. As shown in Fig. 2, (Papyan et al., 2020) characterize it by four manifestations in the classifier and last-layer features: (NC1) the within-class variation collapses to zero; (NC2) the class means converge to simplex Equiangular Tight Frame; (NC3) the class means and the weights of linear

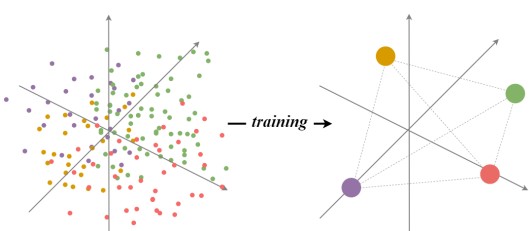

Figure 2: Illustration of Neural Collapse.

classifiers converge to each other; (NC4) the classifier converges to the nearest class-center classifier. Under the constraint of cross-entropy loss, (NC3) and (NC4) occur simultaneously with (NC1) and (NC2). There is considerable research on Neural Collapse (Zhu et al., 2021; Li et al., 2022; Baek et al., 2022; Papyan et al., 2020; Tirer & Bruna, 2022), they mostly directly observe NC using (NC1) (Li et al., 2022; Zhu et al., 2021) because in well-trained models, (NC2) occurs simultaneously with (NC1). Actually, these manifestations suggest models are learning maximally separable features between classes, which can be simplified as three properties: between-class separability, within-class compactness, and the equiangularity between each pair of class distributions. Some previous works Kothapalli (2023); Galanti et al. (2022) explore the impact of the neural collapse phenomenon on generalization and transfer learning. They investigate the impact of pre-trained networks undergoing neural collapse on source training data on various types of target datasets, and obtain various interesting conclusions. However, our task differs from theirs. We aim to find a metric to estimate the transferability of any pre-trained models.

## 3 METHOD

### 3.1 PROBLEM SETUP

We consider a $K$-way classification task on target dataset $D = \{(x_i, y_i)\}_{i=1}^n$, with a total of $n$ labeled samples, and there are $n_k$ samples in the k-th class. Given a pre-trained model zoo $\{\phi_m\}_{m=1}^M$ with a total of $M$ pre-trained models, our goal is to determine a metric score $S_m$ for each model $\phi_m$, and the scores $\{S_m\}_{m=1}^M$ should correlate with their ground truth accuracy which is defined by the test accuracy after fine-tuning.

### 3.2 FAIR COLLAPSE

Inspired by the three properties of the Neural Collapse (NC) phenomenon (Papyan et al., 2020), we propose Fair Collapse (FaCe), which considers three aspects of the target feature spaces: 1) separation between classes; 2) compactness within each class; and 3) class fairness of the model towards target classes. Specifically, the FaCe score $S$ consists of two terms, variance collapse term $C$, corresponding to the first two aspects, and class fairness term $F$, corresponding to the last aspect. Due to the presence of different units of measurement, it is necessary to normalize $C$ and $F$ before adding them together. In summary, for m-th pre-trained model $\phi_m$, FaCe score $S_m$ is formulated as,

$$S_m = \tilde{C}_m + \tilde{F}_m, \quad \tilde{C}_m = \frac{C_m - C_{min}}{C_{max} - C_{min}}, \quad \tilde{F}_m = \frac{F_m - F_{min}}{F_{max} - F_{min}}, \tag{1}$$

where $C_m$ and $F_m$ are the variance collapse and class fairness score of the m-th pre-trained model $\phi_m$, respectively. $\{C_m\}_{m=1}^M$ and $\{F_m\}_{m=1}^M$ are obtained from M pre-trained models and $F_{max}$

and $C_{max}$ are the maximum scores. $F_{min}$ and $C_{min}$ are the minimum scores in $\{F_m\}_{m=1}^M$ and $\{C_m\}_{m=1}^M$. A higher FaCe score $S_m$ indicates that the model's feature space excels in both variance collapse and class fairness, thereby possessing greater transferability. Next, we delve into the details of the variance collapse term and class fairness term.

**Variance Collapse.** This term considers the overall separability of features from different classes. In brief, in the features space of the model with high transferability, features within the same class should be compact, while features between different classes should be far apart. It also measures the gap between unseen source data and the downstream target data. If the gap is small, the source model (*i.e.*, the pre-trained model) should also have a highly separable feature space on the target data. Similar to some works in Neural Collapse studies (Zhu et al., 2021; Li et al., 2022), we simultaneously consider the within-class compactness and the between-class separation by using the magnitude of the between-class covariance compared to within-class covariance. Specifically, for each model, we first calculate the last-layer feature $\boldsymbol{h}_i$ for each target sample $(x_i, y_i)$. Given the global mean $\boldsymbol{h}_G = \frac{1}{n} \sum_{i=1}^n \boldsymbol{h}_i$ and the class mean $\overline{\boldsymbol{h}}_k = \frac{1}{n_k} \sum_{i=1}^n \mathbb{1}(y_i = k)\boldsymbol{h}_i$, where $\mathbb{1}(\cdot)$ denotes the indicator function, the variance collapse score $C$ is formulated as,

$$C = -\frac{1}{K} \operatorname{trace}\left(\boldsymbol{\Sigma}_W \boldsymbol{\Sigma}_B^{\dagger}\right), \tag{2}$$

where $K$ is the number of classes. $\boldsymbol{\Sigma}_W$ is the within-class covariance and $\boldsymbol{\Sigma}_B^{\dagger}$ is the pseudo inverse of between-class covariance $\boldsymbol{\Sigma}_B$. The within-class covariance $\boldsymbol{\Sigma}_W$ and between-class covariance $\boldsymbol{\Sigma}_B$ are computed as,

$$\boldsymbol{\Sigma}_W = \frac{1}{K} \sum_{k=1}^K \sum_{i=1}^n \frac{1}{n_k} \mathbb{1}(y_i = k) \left(\boldsymbol{h}_i - \overline{\boldsymbol{h}}_k\right) \left(\boldsymbol{h}_i - \overline{\boldsymbol{h}}_k\right)^\top, \boldsymbol{\Sigma}_B = \frac{1}{K} \sum_{k=1}^K \left(\overline{\boldsymbol{h}}_k - \boldsymbol{h}_G\right) \left(\overline{\boldsymbol{h}}_k - \boldsymbol{h}_G\right)^\top. \tag{3}$$

A model with a larger $C$ score indicates that its feature space on the target data has a larger between-class distance and a smaller within-class distance. In other words, a larger $C$ signifies better class separability for the corresponding pre-trained model.

**Class Fairness.** In fact, the variance collapse term is usually explicitly or implicitly considered in previous many works (Bao et al., 2019; Pándy et al., 2022; Thakur et al., 2022). However, the equiangularity property of NC, *i.e.*, equal-sized angles between each pair of classes, is usually neglected. In practice, pretrained models can exhibit biases towards specific classes, and relying solely on the variance collapse term does not account for this phenomenon. Take an intuitive example in Fig. 3, there are two types of feature spaces with similar variance collapse scores. Relying solely on the variance collapse term might lead to the selection of the model corresponding to (a) as the best choice. However, this model exhibits bias towards the purple class, which can be detri-

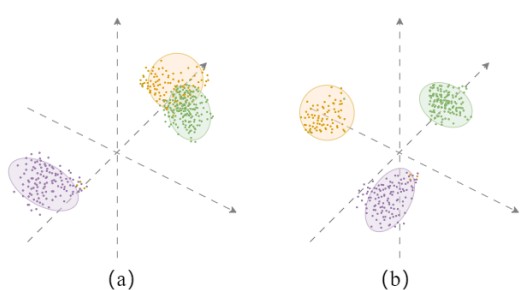

(a)           (b)

Figure 3: Two types of feature spaces with similar variance collapse score. Each point represents a sample in the feature space, with three different colors representing three distinct classes.

mental to the model's training. Note that measuring the equiangularity of class means or classifier weights directly as existing literature Zhu et al. (2021) may be not suitable for our task. This is because the pre-trained model exhibits large intra-class variances on the target data, and the class means can not represent the entire class effectively. Additionally, the classifier dimensions of the pre-trained model differ from the number of classes in the target dataset. As a result, we further consider class fairness, inspired by the equiangularity property in NC, to avoid this issue. Specifically, to better depict the relationships between different classes, we first model each target class as a Gaussian distribution $\mathcal{N}(\overline{\boldsymbol{h}}_k, \boldsymbol{\Sigma}_k)$. $\boldsymbol{\Sigma}_k$ is the within-class covariance, which is defined as,

$$\boldsymbol{\Sigma}_k = \frac{1}{n_k} \sum_{i=1}^n \mathbb{1}(y_i = k) \left(\boldsymbol{h}_i - \overline{\boldsymbol{h}}_k\right) \left(\boldsymbol{h}_i - \overline{\boldsymbol{h}}_k\right)^\top, \tag{4}$$

where $\overline{\boldsymbol{h}}_k$ is the k-th class mean defined in Eq. (3). Specifically, we calculate the overlaps between each pair of class distributions by using the Bhattacharyya coefficient, which has a closed-form

solution when applied between Gaussian distributions. Bhattacharyya distance $D$ between class $k_i$ and $k_j$ is calculated as follows,

$$D\left(k_i, k_j\right) = \tfrac{1}{8}\left(\overline{\boldsymbol{h}}_{k_i} - \overline{\boldsymbol{h}}_{k_j}\right)^{\top} \boldsymbol{\Sigma}^{-1} \left(\overline{\boldsymbol{h}}_{k_i} - \overline{\boldsymbol{h}}_{k_j}\right) + \tfrac{1}{2}\ln\frac{|\boldsymbol{\Sigma}|}{\sqrt{|\boldsymbol{\Sigma}_{k_i}||\boldsymbol{\Sigma}_{k_j}|}}, \tag{5}$$

where $\boldsymbol{\Sigma} = \tfrac{1}{2}\left(\boldsymbol{\Sigma}_{k_i} + \boldsymbol{\Sigma}_{k_j}\right)$. Then, the Bhattacharyya coefficient is defined as $B\left(k_i, k_j\right) = exp - D\left(k_i, k_j\right)$, which indicates the overlap between different class distributions. We can thus obtain the overlap matrix $B$. To highlight the difference between nearby classes and far-away classes, we first convert the overlaps between classes into a probabilistic distribution $P$ by using temperature scaling and softmax function. Then, we calculate the entropy for each row of the overlap matrix and define the class fairness score $F$ as,

$$F = -\frac{1}{K}\sum_{i=1}^{K}\sum_{j=1}^{K} P_{ij}\log P_{ij}, \quad P_{ij} = \frac{\exp\left(B\left(k_i, k_j\right)/t\right)}{\sum_{j'}\exp\left(B\left(k_i, k_{j'}\right)/t\right)}, \tag{6}$$

where $t$ is the temperature in softmax function. Note that, when each row of $P$ approaches a uniform distribution, the class fairness score $F$ reaches its maximum value, which indicates that any class distribution has a similar overlap with the distributions of other classes. From the perspective of Neural Collapse, larger $F$ indicates that the class distributions are closer to various vertices of the simplex Equiangular Tight Frame. From the perspective of model fairness, it suggests that the pre-trained model is fair and exhibits no bias towards specific classes.

## 4 EXPERIMENTS

**Baseline Methods.** In all the experiments, we compare our method with several state-of-the-art methods of various types[2]: LEEP (Nguyen et al., 2020) and NCE (Tran et al., 2019), which are based on the joint distribution of source and target; H-score (Bao et al., 2019) and GBC (Pándy et al., 2022), which are based on the class separability; LogME (You et al., 2021), which is based on the maximum value of label evidence.

**Metric.** The coefficient between our metric and the fine-tuned accuracy is measured by weighted Kendall rank correlation $\tau_w$ (You et al., 2021), which is usually used to measure non-linear, hierarchical, or sequential relationships, and Pearson correlation $\rho$ (Wright, 1921), which is used for measuring linear relationships.

### 4.1 IMAGE CLASSIFICATION: HETEROGENEOUS MODEL ZOO WITH A SINGLE SOURCE

**Experiment Setup.** We construct a model zoo with 15 models pre-trained on ImageNet (Deng et al., 2009) across 5 architecture families: ResNet50, ResNet101, ResNet152 (He et al., 2016), DenseNet121, DenseNet169, DenseNet201 (Huang et al., 2017), MobileNetV1 (Howard et al., 2017), MobileNetV2 (Sandler et al., 2018), MobileNetV3 (Howard et al., 2019), EfficientNetB0, EfficientNetB1, EfficientNetB2, EfficientNetB3 (Tan & Le, 2019), Vgg16, and Vgg19 (Simonyan & Zisserman, 2015). These pre-trained models are directly provided by Pytorch Model Hub[3]. We use 7 standard image classification datasets as the target datasets: basic image recognition datasets CIFAR-10 (Krizhevsky, 2009) and CIFAR-100 (Krizhevsky, 2009); animal dataset Oxford Pets (Parkhi et al., 2012) and CUB (Wah et al., 2011); traffic sign dataset GTSRB (Houben et al., 2013); and describable textures dataset DTD (Cimpoi et al., 2014).

**Training Details.** For the fine-tuning of pre-trained models with different target datasets, We follow the official partitioning to split the training set and validation set. For those without a validation set, we randomly select 10% of the samples to serve as the validation set. Specifically, we use learning rate 1e-2, which always achieves the best accuracy of the validation set in each downstream task, this is sufficient to obtain good transferred models. This protocol is the same as LEEP (Nguyen et al., 2020), and the source selection task in GBC (Pándy et al., 2022). The temperature $t$ in Eq. (6) is empirically set to 0.05. Our experiments are conducted using the PyTorch framework on a 24G NVIDIA Geforce RTX 3090 GPU, and the results are the average of seed 0, 1, 2.

---

[2]LEEP, NCE, LogME: github.com/thuml/LogME; H-score: git.io/J1WOr; GBC is implemented by us.

[3]pytorch.org/hub

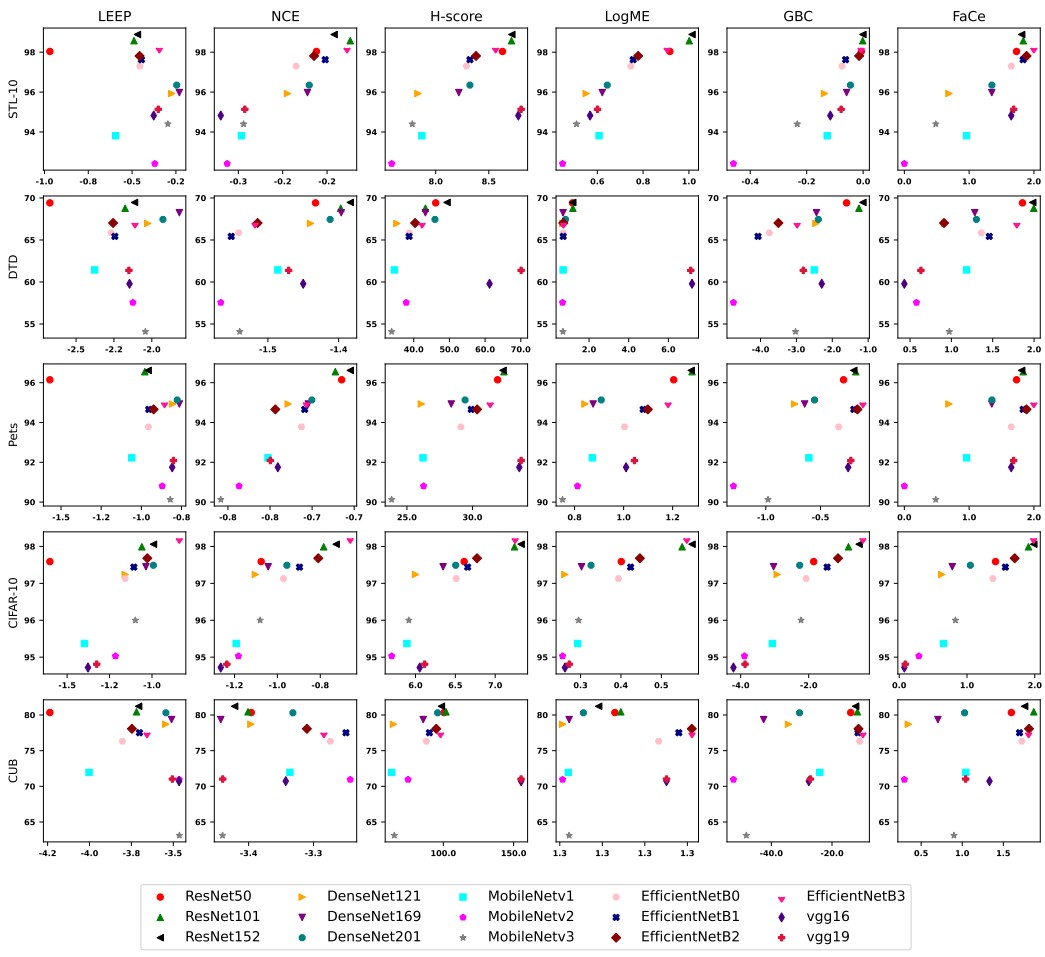

Figure 4: Qualitative results on the heterogeneous model zoo with a single source. For five various datasets, we show the visualized correlation between the accuracy of the fine-tuned model (Y-axis) and the transferability scores (X-axis) of LEEP, NCE, H-score, LogME, GBC, and FaCe.

**Results.** We present the quantitative results in Table 1. The proposed FaCe archives the highest average performance across the seven datasets with $\tau_w = 0.56$ and $\rho = 0.62$. Among these datasets, we have the highest $\tau_w$ on CIFAR-10, CUB, DTD, and STL-10, and the highest $\rho$ on CIFAR-10, DTD, and STL-10. Joint distribution-based method NCE archives the same average $\rho$ as FaCe, and the highest $\tau_w$ on CIFAR-10, the highest $\rho$ on Oxford Pets and STL-10. Furthermore, GBC also yields competitive results. GBC is the summation of between-class overlap, sharing some similarities in motivation with FaCe. However, FaCe additionally considers class fairness, resulting in superior performance compared to GBC.

Table 1: Heterogeneous model zoo with a single source. **Bold** is the best result, underline is the second-best.

| | Target | Method | | | | | |
|---|---|---|---|---|---|---|---|
| | | LEEP | NCE | LogME | H-score | GBC | FaCe |
| **Kendall ($\tau_w$)** | CIFAR-10 | 0.62 | **0.81** | 0.75 | 0.71 | 0.79 | **0.81** |
| | CIFAR-100 | 0.70 | 0.85 | 0.52 | 0.60 | **0.89** | 0.83 |
| | Pet | -0.12 | **0.82** | 0.57 | 0.32 | 0.34 | 0.39 |
| | CUB | -0.34 | -0.19 | 0.06 | 0.23 | 0.23 | **0.33** |
| | GTSRB | **0.20** | 0.07 | -0.37 | 0.10 | -0.05 | 0.10 |
| | DTD | -0.02 | 0.54 | 0.29 | 0.46 | 0.52 | **0.56** |
| | STL-10 | -0.24 | 0.83 | 0.87 | 0.54 | 0.83 | **0.90** |
| | Avg. | 0.11 | 0.53 | 0.38 | 0.42 | 0.51 | **0.56** |
| **Pearson ($\rho$)** | CIFAR-10 | 0.57 | 0.87 | 0.76 | 0.82 | 0.87 | **0.89** |
| | CIFAR-100 | 0.69 | 0.89 | 0.62 | 0.56 | **0.93** | 0.85 |
| | Pet | -0.34 | **0.93** | 0.71 | 0.45 | 0.59 | 0.64 |
| | CUB | -0.38 | 0.03 | 0.12 | -0.02 | **0.57** | 0.39 |
| | GTSRB | **0.28** | 0.15 | -0.71 | 0.18 | 0.00 | -0.05 |
| | DTD | -0.10 | 0.58 | -0.31 | 0.01 | 0.48 | **0.71** |
| | STL-10 | -0.30 | **0.92** | 0.91 | 0.65 | 0.85 | 0.91 |
| | Avg. | 0.06 | **0.62** | 0.30 | 0.38 | 0.61 | **0.62** |

Table 2: Heterogeneous model zoo with multiple sources.

| Target | Method | | | | | |
|---|---|---|---|---|---|---|
| | LEEP | NCE | LogME | H-score | GBC | FaCe |
| **Kendall ($\tau_w$)** | | | | | | |
| DTD | 0.09 | 0.44 | 0.41 | 0.21 | 0.52 | **0.66** |
| Pet | 0.40 | 0.48 | 0.62 | 0.62 | 0.59 | **0.63** |
| STL-10 | 0.36 | 0.41 | 0.54 | 0.39 | 0.52 | **0.57** |
| Avg. | 0.28 | 0.44 | 0.52 | 0.40 | 0.54 | **0.62** |
| **Pearson ($\rho$)** | | | | | | |
| DTD | 0.34 | 0.63 | 0.16 | -0.07 | 0.78 | **0.90** |
| Pet | 0.44 | 0.49 | 0.57 | 0.63 | **0.82** | 0.61 |
| STL-10 | 0.35 | 0.52 | 0.67 | 0.62 | 0.71 | **0.84** |
| Avg. | 0.37 | 0.55 | 0.47 | 0.39 | 0.77 | **0.78** |

Table 3: Homogeneous model zoo with multiple sources and loss functions.

| Target | Method | | | | | |
|---|---|---|---|---|---|---|
| | LEEP | NCE | LogME | H-score | GBC | FaCe |
| **Kendall ($\tau_w$)** | | | | | | |
| DTD | -0.13 | 0.37 | **0.65** | 0.02 | 0.33 | 0.53 |
| STL-10 | -0.40 | -0.25 | 0.42 | **0.63** | 0.04 | 0.58 |
| CIFAR-100 | -0.20 | 0.05 | **0.29** | 0.27 | 0.19 | 0.02 |
| Avg. | -0.24 | 0.06 | **0.46** | 0.31 | 0.19 | 0.38 |
| **Pearson ($\rho$)** | | | | | | |
| DTD | 0.14 | 0.74 | **0.93** | 0.17 | 0.35 | 0.70 |
| STL-10 | -0.78 | -0.54 | 0.44 | **0.71** | -0.10 | 0.68 |
| CIFAR-100 | -0.74 | -0.11 | -0.05 | -0.02 | **-0.01** | -0.30 |
| Avg. | -0.46 | 0.03 | **0.44** | 0.29 | 0.08 | 0.37 |

We show the qualitative results in Fig. 4, *i.e.*, correlation scatter figure between the fine-tuned accuracy and the transferability scores of the comparison methods, where the X-axis is the fine-tuned accuracy, the Y-axis is the transferability score. Pre-trained models with higher fine-tuned accuracy should have higher transferability scores. Therefore, methods where the scatter plot shows an increasing trend are considered superior. We do not achieve the best results in individual experiments, but we still exhibit an obvious increasing trend.

## 4.2 IMAGE CLASSIFICATION: HETEROGENEOUS MODEL ZOO WITH MULTIPLE SOURCES

**Experiment Setup.** We construct a more complex model zoo in this experiment. Specifically, there are a total of 30 heterogeneous pre-trained models from 3 similar magnitude architectures (ResNet50, DenseNet121, and EfficientNetB2) pre-trained on 10 source datasets (CIFAR-10 (Krizhevsky, 2009), CIFAR-100 (Krizhevsky, 2009), CUB (Wah et al., 2011), Oxford Flowers (Nilsback & Zisserman, 2006), Stanford Cars (Krause et al., 2013), Country211 (Radford et al., 2021), Food101 (Bossard et al., 2014), SVHN (Netzer et al., 2011), FGVC Aircraft (Maji et al., 2013)). These datasets encompass a wide range of image types, including animals, plants, digits, food, street, transportation, etc. We conduct the experiments on three benchmark target datasets DTD (Cimpoi et al., 2014), Oxford Pets (Parkhi et al., 2012), and STL-10 (Coates et al., 2011). For the training of pre-trained models on different source datasets, we train the ImageNet model for 100 epochs, using an SGD optimizer with a learning rate of 0.01, and a batch size of 64. The fine-tuned models are obtained under the best hyperparameters.

**Results.** The results are presented in Table 2, the proposed FaCe has the top performance on the average $\tau_w$ and $\rho$, and achieves the best result on two of the three target datasets. Compared to its superior performance in single-source scenarios, NCE appears somewhat less effective in multi-source situations. In contrast, class separability-based method GBC continues to achieve highly competitive results. We speculate that it is inaccurate to use the joint distribution of classifier outputs to estimate the gap between source and target domains in the complex model zoo. Conversely, high-dimension feature-based methods leverage richer information, resulting in superior performance.

## 4.3 IMAGE CLASSIFICATION: HOMOGENEOUS MODEL ZOO WITH MULTIPLE SOURCES AND LOSS FUNCTIONS

**Experiment Setup.** We also construct a homogeneous model zoo, to comprehensively assess the capability of our method. There are a total of 21 ResNet50 models pretrained on 3 source datasets (CIFAR-10 (Krizhevsky, 2009), Oxford Pets (Parkhi et al., 2012) and CUB (Wah et al., 2011)) with 7 widely-employed loss functions [4] (cross entropy (Cover, 1999), label smoothing (Müller et al., 2019), MixUp (Zhang et al., 2018), CutMix (Yun et al., 2019), Cutout (DeVries & Taylor, 2017), large margin softmax cross entropy (Liu et al., 2016), and Taylor softmax cross entropy (Banerjee et al., 2020)). We conduct the experiments on target datasets DTD (Cimpoi et al., 2014), STL-10 (Coates et al., 2011), and CIFAR-100 (Krizhevsky, 2009). The training details of the pre-trained model and fine-tuned model are the same as the setting in Section 4.2 and 4.1, respectively.

---

[4]github.com/fastai/fastai

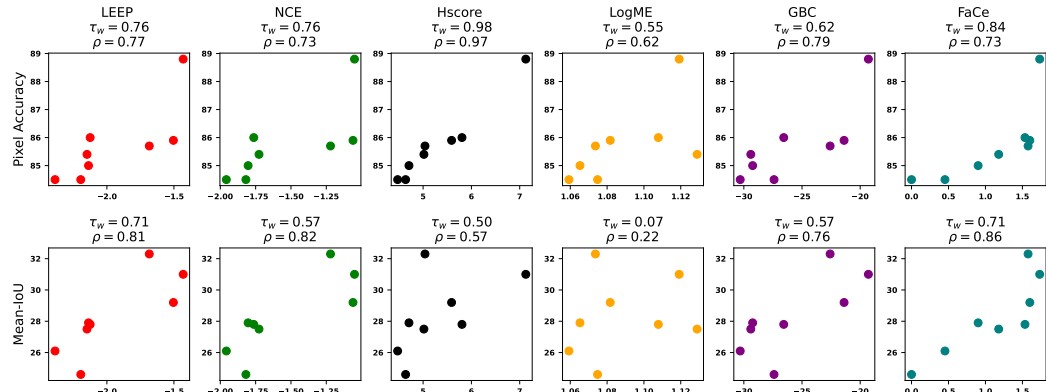

Figure 5: Quantitative and qualitative results on semantic segmentation model zoo.

**Results.** The results are shown in Table 3. In the homogeneous model zoo, half of the methods are ineffective. LogME achieves the highest performance, and FaCe is the second-best. We observe that the performance gap between our method and LogME is marginal. This indicates that our method approaches the state-of-the-art level in estimating the transferability of the homogeneous model zoo.

## 4.4 SEMANTIC SEGMENTATION

**Experiment Setup.** To validate the generalizability of our method, we also conduct experiments in the semantic segmentation scenario. We train 8 models on PSPNet (Zhao et al., 2017) with ResNet50 backbone to construct our segmentation model zoo. These models are trained on 8 different source datasets: ADE20K (Zhou et al., 2017), VOC (Everingham et al., 2012), VOC Aug (Everingham et al., 2012), SBU shadow (Vicente et al., 2016), MSCOCO (Lin et al., 2014), LIP (Gong et al., 2017), kitti (Geiger et al., 2012), and Camvid (Brostow et al., 2009). We compare our method with the state-of-the-art methods on the standard segmentation target dataset CityScapes (Cordts et al., 2016). Following an open-source segmentation benchmark [5], in the pre-training stage, we use SGD optimizer with a learning rate of 1e-4, momentum of 0.9, and WarmupPolyLR scheduler. The training epoch is set to 60, the batch size is 8. To obtain the fine-tuned pixel accuracy and mean IoU, we fine-tune these models on the Cityscapes dataset with the best hyperparameters.

**Results.** We present both quantitative and qualitative results in Fig. 5. In the scenario of pixel accuracy, most of these methods have a satisfactory result, while in the scenario of mean IoU, the performance of these methods has an obvious drop. Pixel accuracy is a metric on the pixel classification problem, while semantic segmentation is essentially a dense prediction problem. FaCe obtains competitive results on pixel accuracy, and the best results on mean IoU. LogME fails in mean IoU, while NCE and H-score also have a certain degree of decline. LEEP, GBC, and FaCe yield similar results under both metrics, demonstrating the generalizability of these three methods in segmentation tasks.

## 4.5 TEXT CLASSIFICATION

**Experiment Setup.** To validate the effectiveness of FaCe on other modalities, we conduct experiments on the Chinese text classification model zoo. We train 6 language models on various architectures and loss functions: NEZHA (Wei et al., 2019), Roberta (Liu et al., 2019), and Roberta with highway (Srivastava et al., 2015), multidrop (Srivastava et al., 2014), Rdrop (Srivastava et al., 2014), and poly loss (Leng et al., 2022). We pre-train these heterogeneous models on source dataset IFLYTEK (Xu et al., 2020), which consists of over 17,000 annotated long-text descriptions related to various app themes relevant to daily life. It encompasses 119 different classes. To obtain the fine-tuned accuracy, we fine-tune these pre-trained models on target dataset TNEWS (Xu et al., 2020), which is derived from the news section of Today's Headlines and comprises news articles from

---

[5]github.com/Tramac/awesome-semantic-segmentation-pytorch

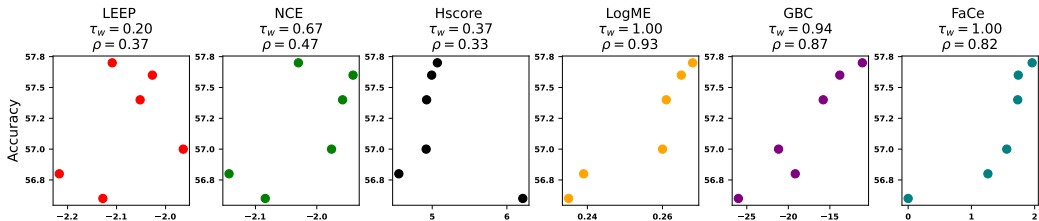

Figure 6: Quantitative and qualitative results on text classification model zoo.

Table 4: Ablation study on (a) heterogeneous model zoo with a single source, (b) heterogeneous model zoo with multiple sources, and (c) homogeneous model zoo with multiple sources and loss functions. The values in the table are the average results on the target datasets.

|      |          | $C$  | $F$  | FaCe |      |          | $C$  | $F$  | FaCe |      |          | $C$  | $F$  | FaCe |
| ---- | -------- | ---- | ---- | ---- | ---- | -------- | ---- | ---- | ---- | ---- | -------- | ---- | ---- | ---- |
| (a)  | $\tau_w$ | 0.49 | 0.56 | 0.56 | (b)  | $\tau_w$ | 0.59 | 0.52 | 0.62 | (c)  | $\tau_w$ | 0.36 | 0.37 | 0.38 |
|      | $\rho$   | 0.52 | 0.61 | 0.62 |      | $\rho$   | 0.77 | 0.56 | 0.78 |      | $\rho$   | 0.27 | 0.37 | 0.36 |

15 different categories, including travel, education, finance, military, and more. We use the same training hyperparameters in a text classification benchmark [6] for both source model pre-training and target model fine-tuning, where epoch is 10, batch size is 16, AdamW (Loshchilov & Hutter, 2017) optimizer with a learning rate 2e-5.

**Results.** Both the quantitative and qualitative results are presented in Fig. 6. LogME, GBC, and FaCe achieve competitive results. Among them, LogME and FaCe can be considered as the optimal solutions of this model zoo since both of them achieve a Kendall rank correlation coefficient $\tau_w$ of 1, which is the best result attainable. This demonstrates the generalizability and effectiveness of FaCe in the text modality.

## 4.6 ABLATION STUDY

We validate the effectiveness of the two terms in FaCe, *i.e.*, variance collapse ($C$) term and class fairness ($F$) term on the three types of model zoos described in Section 4.1, 4.2, and 4.3. In Table 4, we provide results using only $C$ and $F$ separately, and the results using complete FaCe. The ablation experiments reveal that the effectiveness of FaCe's two terms varies across different tasks. For instance, in (a) a heterogeneous model zoo with a single source, the CF component yields better results, while in (b) a heterogeneous model zoo with multiple sources, the opposite is true. $C$ measures the within-class compactness and between-class separation in the feature space, while $F$ measures the uniformity of class distributions in the feature space. These two terms provide different perspectives on the feature space structure metric. FaCe combines the strengths of both, resulting in the best overall performance.

## 5 CONCLUSION

In this paper, we study the transferability estimation problem and propose a novel metric Fair Collapse (FaCe) which is motivated by the Neural Collapse (NC) phenomenon. Specifically, we investigate the Neural Collapse of pre-trained models and their fine-tuned models and observe a strong correlation between the NC of the fine-tuned models and the corresponding pre-trained models. Inspired by this observation, we introduce FaCe to estimate the transferability from two perspectives, *i.e.*, variance collapse and class fairness. Our class fairness term in FaCe considers the bias of the pre-trained model towards specific classes, addressing an issue that has been neglected in prior research. Fair Collapse serves as an application of the Neural Collapse phenomenon in the context of transferability estimation tasks, and we aspire that our work can shed some light on the community.

---

[6]github.com/shawroad/Text-Classification-Pytorch

## 6 REPRODUCIBILITY STATEMENT

To ensure the reproducibility of our method, we provide the pseudo-code in the Appendix, and all the hyper-parameters are given in Section 4. The image classification benchmark in Section 4.1, 4.2, and 4.3 is implemented by us. The semantic segmentation [7]. and text classification benchmarks[8] are implemented based on open-source repositories. We provide the code of our image classification benchmark and FaCe metric in the supplementary zip file.

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

## A    APPENDIX

### A.1    ALGORITHM

---

**Algorithm 1** Algorithm of the proposed FaCe.

---

**Input:** A Model Zoo $\{\phi_m\}_{m=1}^M$ with M pre-trained models; target dataset $D = \{(x_i, y_i)\}_{i=1}^n$, with a total of $n$ labeled samples, and there are $n_k$ samples in the k-th class;

1: **repeat**
2:     Given a pre-trained model $\phi_m$, obtain the last-layer features $\{h_i\}_{i=1}^n$ on $D = \{(x_i, y_i)\}_{i=1}^n$;
3:     Calculate the variance collapse score $C_m$ for $\phi_m$ by Eq. (2);
4:     Calculate the class fairness score $F_m$ for $\phi_m$ by Eq. (4);
5: **until** Obtain $\{C_m\}_{m=1}^M$ and $\{F_m\}_{m=1}^M$ for $\{\phi_m\}_{m=1}^M$;
6: Rescale $\{C_m\}_{m=1}^M$ and $\{F_m\}_{m=1}^M$, and obtain the FaCe score $\{S_m\}_{m=1}^M$ by Eq. (1).
**Output:** Transferability ranking of pre-trained models.

---

### A.2    DISCUSSION

FaCe is a method inspired by Neural Collapse, and the class fairness term is a variant of the equiangularity in Neural Collapse. A solution (Zhu et al., 2021; Papyan et al., 2020) to estimate the equiangularity is to quantify the closeness of the classifier weights to a simplex Equiangular Tight Frame (ETF) directly: $\mathcal{NC}_2(\boldsymbol{W}) = \left\| \frac{\boldsymbol{WW}^\top}{\|\boldsymbol{WW}^\top\|_F} - \frac{1}{\sqrt{K-1}} \left( \boldsymbol{I}_K - \frac{1}{K}\boldsymbol{1}_K\boldsymbol{1}_K^\top \right) \right\|_F$, where $\boldsymbol{W} \in \mathbb{R}^{K \times d}$ is the weight of the classifier. In our task, this is actually an equiangularity metric for the unknown source dataset rather than the target dataset, since the model is pre-trained on the source dataset. Due to the self-duality between model weights and class means, a naive solution is to replace $\boldsymbol{W}$ in the above equation with target class means

Table 5: Comparison between our class fairness term and the naive equiangularity metric.

|  | CF | $\mathcal{NC}_2(\boldsymbol{A})$ |
|---|---|---|
| $\tau_w$ | 0.56 | 0.32 |
| $\rho$ | 0.61 | 0.37 |

Table 6: Heterogeneous model zoo with a single source. **Bold** is the best result, underline is the second-best.

| Target | | LEEP | NCE | LogME | H-score | GBC | FaCe |
|---|---|---|---|---|---|---|---|
| | | | | Method | | | |
| Kendall ($\tau_w$) | CIFAR-10 | 0.62 | **0.81** | 0.75 | 0.71 | 0.79 | 0.81 |
| | CIFAR-100 | 0.70 | 0.85 | 0.52 | 0.60 | **0.89** | 0.83 |
| | Pet | -0.12 | **0.82** | 0.57 | 0.32 | 0.34 | 0.39 |
| | CUB | -0.34 | -0.19 | 0.06 | 0.23 | 0.23 | **0.33** |
| | GTSRB | **0.20** | 0.07 | -0.37 | 0.10 | -0.05 | 0.10 |
| | DTD | -0.02 | 0.54 | 0.29 | 0.46 | 0.52 | **0.56** |
| | STL-10 | -0.24 | 0.83 | 0.87 | 0.54 | 0.83 | **0.90** |
| | Food-101 | 0.22 | 0.54 | 0.26 | 0.28 | **0.60** | 0.58 |
| | Avg. | 0.13 | 0.53 | 0.37 | 0.24 | 0.52 | **0.56** |
| Pearson ($\rho$) | CIFAR-10 | 0.57 | 0.87 | 0.76 | 0.82 | 0.87 | **0.89** |
| | CIFAR-100 | 0.69 | 0.89 | 0.62 | 0.56 | **0.93** | 0.85 |
| | Pet | -0.34 | **0.93** | 0.71 | 0.45 | 0.59 | 0.64 |
| | CUB | -0.38 | 0.03 | 0.12 | -0.02 | **0.57** | 0.39 |
| | GTSRB | **0.28** | 0.15 | -0.71 | 0.18 | 0.00 | -0.05 |
| | DTD | -0.10 | 0.58 | -0.31 | 0.01 | 0.48 | **0.71** |
| | STL-10 | -0.30 | **0.92** | 0.91 | 0.65 | 0.85 | 0.91 |
| | Food-101 | 0.30 | **0.74** | 0.31 | 0.23 | 0.64 | 0.73 |
| | Avg. | 0.09 | **0.64** | 0.30 | 0.36 | 0.62 | 0.63 |

matrix $\boldsymbol{A} \in \mathbb{R}^{K \times d}$. As shown in Table 5, we conduct the comparison experiments of our class fairness term and this naive solution on the heterogeneous model zoo with a single source. CF is our class fairness term, which is obviously superior to $\mathcal{NC}_2(\boldsymbol{A})$. The equiangularity in Neural Collapse essentially implies the maximum separability of class distributions in the feature spaces. When the within-class variance collapses to zero, each class mean can represent the corresponding entire class distribution. In the cases of a pre-trained model without fine-tuning, the within-class variance is large, hence the closeness between the class means and a simplex ETF cannot accurately measure the separability of class distributions.

## A.3 MORE RESULTS

We provide additional results on more target datasets on three image classification model zoos. On the heterogeneous model zoo with single source (corresponding to Table 1), we add a large-scale dataset Food101 Bossard et al. (2014) in Table 6. On the heterogeneous model zoo with multiple sources (corresponding to Table 2), we add STL-10 Coates et al. (2011), Cifar-10 Krizhevsky (2009), Cifar-100 Krizhevsky (2009), and GTSRB Houben et al. (2013) in Table 7. On the homogeneous model zoo with multiple sources and loss functions (corresponding to Table 3), we add Cifar-10 Krizhevsky (2009), Oxford Pets Parkhi et al. (2012), CUB Wah et al. (2011), GTSRB Houben et al. (2013) in Table 8.

Table 7: Heterogeneous model zoo with multiple sources.

| Target | Method | | | | | |
|---|---|---|---|---|---|---|
| | LEEP | NCE | LogME | H-score | GBC | FaCe |
| **Kendall** ($\tau_w$) | | | | | | |
| DTD | 0.09 | 0.44 | 0.41 | 0.21 | 0.52 | **0.66** |
| Pet | 0.40 | 0.48 | 0.62 | 0.62 | 0.59 | **0.63** |
| STL-10 | 0.36 | 0.41 | 0.54 | 0.39 | 0.52 | **0.57** |
| Cifar-10 | 0.06 | 0.41 | 0.32 | 0.28 | **0.45** | 0.40 |
| Cifar-100 | -0.10 | **0.20** | 0.04 | -0.02 | 0.18 | 0.17 |
| GTSRB | 0.15 | **0.16** | -0.16 | -0.10 | -0.05 | -0.05 |
| CUB | 0.43 | 0.45 | **0.66** | 0.37 | 0.62 | 0.56 |
| Avg. | 0.20 | 0.36 | 0.35 | 0.25 | 0.40 | **0.42** |
| **Pearson** ($\rho$) | | | | | | |
| DTD | 0.34 | 0.63 | 0.16 | -0.07 | 0.78 | **0.90** |
| Pet | 0.44 | 0.49 | 0.57 | 0.63 | **0.82** | 0.61 |
| STL-10 | 0.35 | 0.52 | 0.67 | 0.62 | 0.71 | **0.84** |
| Cifar-10 | 0.23 | 0.32 | 0.27 | 0.33 | **0.61** | **0.64** |
| Cifar-100 | -0.01 | **0.05** | -0.03 | -0.03 | **0.14** | 0.09 |
| GTSRB | **0.36** | **0.36** | 0.29 | 0.23 | 0.00 | 0.06 |
| CUB | 0.37 | 0.44 | **0.31** | 0.41 | **0.82** | 0.77 |
| Avg. | 0.30 | 0.40 | 0.32 | 0.30 | 0.55 | **0.56** |

Table 8: Homogeneous model zoo with multiple sources and loss functions.

| Target | Method | | | | | |
|---|---|---|---|---|---|---|
| | LEEP | NCE | LogME | H-score | GBC | FaCe |
| **Kendall** ($\tau_w$) | | | | | | |
| DTD | -0.13 | 0.37 | **0.65** | 0.02 | 0.33 | 0.53 |
| STL-10 | -0.40 | -0.25 | 0.42 | **0.63** | 0.04 | 0.58 |
| CIFAR-100 | -0.20 | 0.05 | **0.29** | 0.27 | 0.19 | 0.02 |
| Cifar-10 | -0.07 | -0.12 | -0.05 | **-0.03** | -0.10 | -0.06 |
| Pet | **0.50** | 0.47 | 0.38 | 0.41 | 0.37 | 0.35 |
| CUB | 0.54 | 0.70 | 0.72 | 0.70 | **0.83** | 0.69 |
| GTSRB | -0.08 | -0.23 | -0.16 | -0.17 | **0.24** | 0.19 |
| Avg. | 0.02 | 0.14 | 0.32 | 0.26 | 0.27 | **0.33** |
| **Pearson** ($\rho$) | | | | | | |
| DTD | 0.14 | 0.74 | **0.93** | 0.17 | 0.35 | 0.70 |
| STL-10 | -0.78 | -0.54 | 0.44 | **0.71** | -0.10 | 0.68 |
| CIFAR-100 | -0.74 | -0.11 | -0.05 | -0.02 | **-0.01** | -0.30 |
| Cifar-10 | -0.03 | 0.00 | -0.11 | -0.01 | **0.08** | 0.04 |
| Pet | 0.52 | 0.65 | 0.52 | **0.76** | 0.74 | 0.67 |
| CUB | 0.62 | 0.61 | 0.45 | 0.62 | **0.83** | 0.72 |
| GTSRB | 0.01 | -0.18 | -0.11 | -0.06 | 0.37 | **0.38** |
| Avg. | -0.04 | 0.17 | 0.30 | 0.31 | 0.32 | **0.41** |

