# OpenReview forum: "Unleashing the power of Neural Collapse for Transferability Estimation"
_ICLR.cc/2024/Conference — Submitted to ICLR 2024_

### Official Review · Reviewer_TZ7d · 2023-11-01

**Soundness:** 2 fair
**Presentation:** 2 fair
**Contribution:** 1 poor
**Rating:** 6
**Confidence:** 4

**Summary:**

The paper studies the notion of transferability of pretrained models, ie how well one can predict the performance of a model on a downstram task after finetuning on the target dataset. The paper proposes using the degree of Neural Collapse  (Papyan et al., 2020) as a way of measuring the transferability of models. The proposed metric, FaCe, is a function of inter-/intra class covariance and distributions.

**Strengths:**

* The paper deals with a hard and open problem: transferabiltiy estimation

* It proposes a solution based on properties of NC of the pretrianed model, and shows some gains vs other transferability metrics.

* the paper seems reproducible, as the authors further shared code as supplementary materials

**Weaknesses:**

1. There is a very relevant related work that is discussing NC for transfer learning, that is currently not cited in this submission:
Galanti, Tomer, András György, and Marcus Hutter. "On the role of neural collapse in transfer learning." ICLR 2022.
The authors should discuss this paper and the contributions of the submission related to that.

2. The authors claim that a key component of their method is the class fairness term. It is unclear how FaCe compares to a variant without that term (and everything else the same), ie if S_m = C_m and not C_m+F_m. There is no ablation to understand how this part affects the final metric.

3. The experimental validation is weak. There are very few and small -scale datasets in Tab1, while even fewer in Tabs 2 and 3. A more extensive evaluation is needed to showcase any improvements of FaCe vs the other metrics. Currently this is not clear from Tab1 where there seems to not be a clear winner at all.


A note:
A very related concurrent work from ICCV 2023 should be cited and possibly also discussed (as concurrent work, of course, not limiting this papers novelty):
Wang, Zijian, et al. "How Far Pre-trained Models Are from Neural Collapse on the Target Dataset Informs their Transferability." Proceedings of the IEEE/CVF International Conference on Computer Vision. 2023.

**Questions:**

- Can you provide ablations for performance when only one of the two terms is used, ie C and F for FaCe? ie add two more columns in tables 1/2/3 with that. I understand that GBC is close, but it would be great to see the performance of your method, in your experimental setup, for each of the two terms separately and then together. I think this would help clarify the contribution of this paper.

- What are only a subset of papers presented in Tab2 and 3? Ca you provide results for all, eg in an appendix if the issue is space?

---

> ### Author Response · Authors · 2023-11-19
>
> We thank the reviewer for providing valuable comments and address your concerns in the following responses.
>
> > **[Q1]**: "On the role of neural collapse in transfer learning. [1]" ICLR 2022. The authors should discuss this paper and the contributions of the submission related to that.
>
>
> **[A1]** Thank you for your kind reminder. We added a discussion in the related work section.
>
>
> [1] investigates the impact of pre-trained networks undergoing neural collapse on source training data on various types of target datasets. The authors find that neural collapse occurring on the source data can generalize to new samples and even unseen categories. We list the main difference from [1] below:
>
> - The task and setting are different.  [1] study the ability of models, which undergo neural collapse on source tasks, to learn representations for classification that are transferable to new, unseen classes. However, our task is to propose a metric to estimate the transferability of a given pre-trained model.  Whether this model undergoes neural collapse on the source data is unknown.
> - [1] primarily investigates the performance of models experiencing neural collapse in experiments to learn about phenomena and draw conclusions. In contrast, our work is inspired by the characteristics of neural collapse to propose a method for transferability estimation.
>
> > **[Q2]**: The authors claim that a key component of their method is the class fairness term. It is unclear how FaCe compares to a variant without that term. There is no ablation to understand how this part affects the final metric.
>
> **[A2]** We agree that the ablation study is crucial. As shown in Section 4.6, we provide the ablation study of two terms of FaCe. The results validate that both terms are effective, and they complement each other to achieve the best average performance.
>
>
> > **[Q3]**: The experimental validation is weak. A more extensive evaluation is needed to showcase any improvements of FaCe vs the other metrics.
>
> **[A3]** We follow existing work and select a set of common-used representative target datasets. These datasets cover various data types, such as animals, traffic signs, textures, and more, providing a diverse set of evaluation scenarios.
>
> **In Appendix A.3, we add experiments on the large-scale dataset food101 (~101,000 images) in Table 1, and include more target datasets in Table 2, 3.**
>
> As shown in Appendix A.3, Table 6,7,8, in the presence of a large-scale dataset, we still achieve competitive average performance (Table 6). Moreover, on more target datasets, we also obtain the best average performance on the other two model zoos (Table 7, 8).
>
>
> > **[Q4]**: A very related concurrent work from ICCV 2023 should be cited and possibly also discussed (as concurrent work, of course, not limiting this paper's novelty):
>
> **[A4]** Thank you for your kind reminder. We have also noticed the paper [2], which was released after our submission and shares similar ideas with ours.  We have added a discussion about it in the related work section.
>
> NCTI (their method) is a concurrent work of our method, which is also inspired by neural collapse. NCTI consists of three terms, which correspond to three characteristics of neural collapse, respectively.  Different from NCTI, which assesses the geometry structure based on nuclear norm, FaCe is based on the class distribution overlapping matrix entropy. We will make effort to reproduce this work and include a comparative analysis with it in the future.
> - [2] How Far Pre-trained Models Are from Neural Collapse on the Target Dataset Informs their Transferability. ICCV 2023.
>
>
> >   **[Q5]**: What is only a subset of papers presented in Tab 2 and 3? Can you provide results for all, eg in an appendix if the issue is space?
>
> **[A5]** In Table 1-3, the comparison methods are the same (LEEP, NCE, LogME, H-score, GBC).
>
> We suspect that when you mentioned "a subset of papers," you might refer to the target datasets. We also provide more results in Appendix A.3. **On these new target datasets, FaCe still achieves state-of-the-art performance.**

---

> ### Author Response · Authors · 2023-11-21
> **Invitation to further discussion**
>
> Dear reviewer,
>
> We genuinely appreciate the time and effort you've invested in reviewing our paper. We have carefully provided relevant responses and results to your concerns. We are eager to further discuss with you and gain your insights **before the end of the Author/Reviewer phase**. Please let us know if any aspect of our work remains unclear or if you have additional feedback.
>
> Thank you.

---

> > ### Comment · Reviewer_TZ7d · 2023-11-22
> > **Thank you for the responses**
> >
> > I want to thank the reviewers for their responses.
> >
> > > addition of missing reference [1]
> >
> > Thank you for discussing this. It is true that the task is different but I see a lot of shared motivation.
> >
> > > Ablation on F and C contributions
> >
> > Thank you for making this ablation now clear. However, the results in Tab 4 suggest that the class fairness term only helps in 1 out of three zoo cases ( homogeneous model zoo with multiple sources ) and not in the other two. Are there any intuitions why? this is the main contribution and it should be thoroughly discussed.
> >
> > >  select a set of common-used representative target datasets.
> >
> > I still belive the evaluation to be not as strong as needed. The set selected is a _subset_ of the datasets commonly used, ie 7 vs 9 used in LogME and 8 in GBC. Conclusions would be much stronger if results on all datasets used in related works were there.
> >
> > I have one more question/clarification on the hyperparameters used. The authors say:
> > >  We used the validation set to find the best hyperparameters
> >
> > Which val set of which dataset is used? Are the same hyperparameters used in all experiments? Is the exact same protocol also used for setting the hyperparameters of the compared methods and baselines?
> >
> > Thank you

---

> > > ### Author Response · Authors · 2023-11-22
> > >
> > > Thanks for further discussion, we would like to address these remaining concerns as follows.
> > >
> > > > It is true that the task is different but I see a lot of shared motivation.
> > >
> > > They study the ability of models undergoing neural collapse on downstream tasks. While our motivation is also related to the phenomenon of neural collapse, we do not simply study the effect, but even propose a new metric to estimate the transferability of a given pre-trained model.
> > >
> > > > The results in Tab 4 suggest that the class fairness term only helps in 1 out of three zoo cases ( homogeneous model zoo with multiple sources ) and not in the other two. Are there any intuitions why? this is the main contribution and it should be thoroughly discussed.
> > >
> > >
> > > The reviewer might have confused C and **F (our key contribution)** in Table 4.
> > >
> > > Note that the Class Fairness term **(F)** works in all three tasks. To be specific, in (a) heterogeneous model zoo with single source, and \(c\) homogeneous model zoo with multiple sources and loss functions, using F alone can approximate the final FaCe results. This clearly illustrates the effectiveness of F.
> > >
> > >
> > > We posit the reason of the variance collapse term (**C**) being less effective, F potentially enlargers the between-class variance, which is some overlap with the role of C. FaCe combines these two aspects and achieves the highest performance.
> > >
> > >
> > > > I still believe the evaluation to be not as strong as needed. The set selected is a subset of the datasets commonly used, ie 7 vs 9 used in LogME and 8 in GBC. Conclusions would be much stronger if results on all datasets used in related works were there.
> > >
> > > We agree that it is fairer to include all the target datasets of the compared methods. However, in this field, different papers typically utilize different downstream datasets on different pre-trained models. Even the two mentioned papers (LogMe and GBC) differ. In the experiments of GBC, datasets like Imagenette, CUB, Dogs, and Flower are not conducted by LogME. Thus, we simply adopt 7 commonly used datasets for comparing transferability estimation methods.
> > >
> > >
> > > In the rebuttal, we further provide results on a new target dataset Food101, and there are a total of 8 datasets. Besides, we also validate the effectiveness of our method on various tasks, including image classification, semantic segmentation, and even text classification. Compared to other works in this field, our experiments are extensive. We believe such extensive results can well support the effectiveness of FaCe in transferability estimation tasks.
> > >
> > >
> > >
> > > > Which val set of which dataset is used? Are the same hyperparameters used in all experiments? Is the exact same protocol also used for setting the hyperparameters of the compared methods and baselines?
> > >
> > > We follow the official partition to obtain the validation set. For those without a validation set, we randomly select 10% of the samples to serve as the validation set.
> > >
> > > Existing methods [LEEP, GBC, LogME] select the best hyper-parameters to ensure fully fine-tuning.
> > >
> > > We also select the best hyper-parameter and fully fine-tune the pre-trained models. Specifically, we define the search space of learning rate [1e-3, 1e-2, 1e-1], and 1e-2 always achieves the best accuracy of the validation set in each downstream task, this is sufficient to obtain good transferred models. This protocol is the same as LEEP, and the source selection task in GBC.
> > >
> > >
> > >
> > > **We hope that these answers address your concerns effectively, and we look forward to your response.**

---

> > > > ### Comment · Reviewer_TZ7d · 2023-11-22
> > > > **Thanks - but one more thing**
> > > >
> > > > Thank you for your quick reply. I agree there exist differences in your work, it is just that this missing reference slightly decreases it.
> > > >
> > > > You are right, I apologize for confusing F and C. The gains seem clear now in T4.
> > > >
> > > > I do however have one remaining question  from your response: are you assuming the existence of a validation set from each _downstream task_ and tune your methods parameters with that, per dataset? Do you consider this set labeled or unlabeled? what would be your method’s performance when you simply use one set of parameters for all downstream datasets? Apologies for any misunderstanding.

---

> > > > > ### Author Response · Authors · 2023-11-22
> > > > >
> > > > > We are glad to have solved most of your concerns.
> > > > >
> > > > > > Are you assuming the existence of a validation set from each downstream task and tune your methods parameters with that, per dataset? Do you consider this set labeled or unlabeled? what would be your method’s performance when you simply use one set of parameters for all downstream datasets?
> > > > >
> > > > > Given pre-trained models, **the transferability estimation task** aims to rank fine-tuned models based on the accuracies on **downstream supervised datasets**. Before estimation, we need to obtain each fine-tuned model and its accuracy for each downstream task.
> > > > >
> > > > > In particular, we follow LEEP to select the same hyperparameter of fine-tuning for all the downstream tasks, namely, the highest accuracy on each labeled validation set is achieved under the same hyperparameter.
> > > > >
> > > > > Note that **the fine-tuned hyperparameters are independent of transferability estimation methods**. For our method, FaCe, the only hyperparameter within FaCe is the temperature $t$, which is fixed for all the downstream tasks in this paper.
> > > > >
> > > > > Hope this could address your concern.

---

> ### Comment · Reviewer_TZ7d · 2023-11-22
> **Thank you**
>
> Alright, thanks for clarifying an important part  of the tuning for the task that I confused. My apologies again, it is busy times.
>
> Based on the author responses I will increase my score to borderline accept.

---

> > ### Author Response · Authors · 2023-11-23
> > **Thank you for your endorsement and the increased score.**
> >
> > We are delighted to receive your response. We are profoundly grateful for your endorsement of our paper.

---

### Official Review · Reviewer_Rkcn · 2023-11-01

**Soundness:** 2 fair
**Presentation:** 3 good
**Contribution:** 2 fair
**Rating:** 5
**Confidence:** 4

**Summary:**

The paper introduces a metric designed to evaluate the transferability of a model trained under the so-called Terminal Phase of Training (TPT), at which neural collapse manifests. The key idea is to use a Gaussian distribution surrounding the collapsed feature. Subsequently, the Bhattacharyya score (available in this case in closed form) is computed to assess the difference between the pretrained model and its fine-tuned version. The authors support their approach with empirical evidence, drawing from multiple datasets and models.

**Strengths:**

The practical result of selecting the optimal model from a set of pretrained models, which best fine-tunes with the available data prior to training, appears to be a significant strength, given its potential appeal to a broad audience and its broader implications. Additionally, the closed-form formulation further shows its strength, with implications both in practical and theoretical contexts.

**Weaknesses:**

Considering the rapid dissemination of models online by third-party sources, the approach of selecting the optimal model for fine-tuning might have constrained applicability.

The assertion of a strong correlation is not clearly evident from the paper. For example It would be beneficial if from the experimental results some takeaway message were more explicit provided. In its current shape the paper requires visiting the experimental result section and understanding which specific model or architecture best transfer according to the proposed metric. Given that a significant portion of the paper's contribution is about empirical validation of the proposed transferability metric, this aspect should be improved. This is important as the paper claims "strong correlation" both in the abstract and conclusion.

The paper frequently refers to the "domain shift" causing features not to lie on the hypersphere. The implications of this assertion are not clear, and what is meant by it being "too strict," needs clearer elaboration. The paper seems to lack a direct discussion on this. In which way not lying on the hypersphere is related to neural collapse? For instance, could the final classifier bias parameters be disregarded, thereby naturally aligning features with the hypersphere? The connection between domain shift and features not aligning with the hypersphere should be better clarified.

The metric introduced appears to hold even when training does not proceed until TPT. Given this, what is the significance of emphasizing the necessity for neural collapse in the proposed method?

**Questions:**

Questions and weaknesses are grouped together to facilitate a clearer understanding and correlation of the issues.

---

> ### Author Response · Authors · 2023-11-19
>
> We thank the reviewer for providing valuable comments and address your concerns in the following responses.
>
> > **[Q1]**: Considering the rapid dissemination of models online by third-party sources, the approach of selecting the optimal model for fine-tuning might have constrained applicability.
>
> **[A1]** It is precisely because there are numerous third-party models that it becomes necessary to evaluate these models to estimate their suitability for downstream tasks without time-consuming fine-tuning for all the models.
>
> > **[Q2]**: The assertion of a strong correlation is not clearly evident from the paper. For example, It would be beneficial if, from the experimental results, some takeaway messages were more explicitly provided.
>
> **[A2]** We feel sorry for any confusion. We want to explain the strong correlation observed in Figure 1 as follows. The NC score ranking in these pre-trained models remains mostly consistent during fine-tuning. For instance, in subfigure (a), the yellow line represents the variation curve in NC for ResNet152 during the fine-tuning process. It starts with the highest NC on the target data from the initial moment and continues to maintain the highest NC throughout fine-tuning. The black curve (DenseNet201) starts with the lowest degree of NC, and continues to maintain the lowest level during the fine-tuning process. Other models also exhibit similar patterns, therefore, we posit the existence of a strong correlation between the NC of the pre-trained model and that of the fine-tuned model.
>
>
>
> > **[Q3]**: In its current shape the paper requires visiting the experimental result section and understanding which specific model or architecture best transfers according to the proposed metric.
>
>
> **[A3]** Our metric does not exhibit bias towards specific network structures or loss functions. For example, on the DTD dataset, FaCe gives ResNet152 the highest score, while on Cifar-10, FaCe gives EfficientNetB3 the highest score.
>
> This sounds reasonable for a qualified transferability estimation score since the best model is related to the specific downstream task. For different tasks, the best model, loss function or source datasets vary.
>
> The task of transferability estimation aims to find a metric for assessing the appropriateness of pre-trained models for downstream tasks, rather than evaluating the effectiveness of particular architectures or loss functions.
>
>
> > **[Q4]**: The paper frequently refers to the "domain shift" causing features not to lie on the hypersphere. The implications of this assertion are not clear, and what is meant by it being "too strict," needs clearer elaboration. The paper seems to lack a direct discussion on this.
>
> **[A4]** When Neural Collapse (NC) occurs, the features of the training data collapse to class means, and the class means, along with the classifier weights, converge to the vertices of a simplex equiangular tight frame (ETF). At this point, the class means of the features can be considered as distributed on a hypersphere [1]. However, the model is pre-trained on the source data. On downstream target data,  domain shift exists, and the class means will not be located on the hypersphere.
>
> Our class fairness term is inspired by NC2. A widely-used metric in existing literatures is defined as $$ \mathcal{NC}_2(W) = \left\|\frac{W W^{\top}}{\left\|{W} {W}^{\top}\right\|_F}-\frac{1}{\sqrt{K-1}}\left({I}_K-\frac{1}{K} \mathbf{1}_K \mathbf{1}_K^{\top}\right)\right\|_F.$$ Essentially, this metric measures the equiangularity of the classifier weights or class means. To use this metric, it should be under the scenarios where the class means are located on the hypersphere. This is the meaning of "too strict" in our manuscript.
>
>
> We apologize for any confusion caused by the unclear description.  We have clarified the relevant description in Section 3.2.
> - [1] Neural Collapse: A Review on Modelling Principles and Generalization. TMLR 2023.
>
>
> > **[Q5]**: Could the final classifier bias parameters be disregarded, thereby naturally aligning features with the hypersphere?
>
> **[A5]** The classifier is pre-trained on the source dataset, which has a different label space from the downstream target dataset, therefore, using the classifier makes no sense.
>
>
> > **[Q6]**: The metric introduced appears to hold even when training does not proceed until TPT. Given this, what is the significance of emphasizing the necessity for neural collapse in the proposed method?
>
> **[A6]** Sorry for the misunderstanding. We do not assume Neural Collapse in fine-tuning the pre-trained model. By contrast, we study the degree of NC of pre-trained models during fine-tuning, through the process and find an interesting observation. Our method is motivated by this observation, to estimate the degree of NC of the pre-trained model, and it is independent of whether training has reached TPT.

---

> ### Author Response · Authors · 2023-11-21
> **Invitation to further discussion**
>
> Dear reviewer,
>
> We genuinely appreciate the time and effort you've invested in reviewing our paper. We have carefully provided relevant responses and results to your concerns. We are eager to further discuss with you and gain your insights **before the end of the Author/Reviewer phase**. Please let us know if any aspect of our work remains unclear or if you have additional feedback.
>
> Thank you.

---

> ### Author Response · Authors · 2023-11-23
>
> Dear Reviewer,
>
> Since the discussion deadline is approaching in less than 11 hours, we kindly request your feedback on whether the response adequately addresses your concerns. If you have any more questions, we would be happy to provide further clarification.
>
> Your timely response is greatly appreciated.
>
> Thanks.

---

### Official Review · Reviewer_JPai · 2023-11-02

**Soundness:** 3 good
**Presentation:** 3 good
**Contribution:** 3 good
**Rating:** 6
**Confidence:** 3

**Summary:**

The "pre-training followed by fine-tuning" paradigm has become the standard training approach for many tasks in the deep learning domain. Selecting the appropriate pre-trained model for a specific downstream task poses a challenge. This paper introduces a transferability estimation method, Fair Collapse (FaCe). The method aims to determine a metric that indicates the potential performance of pre-trained models on a target dataset without the need to fine-tune every model. Ideally, transferability metrics should closely correlate with the actual performance post-fine-tuning.

**Strengths:**

1.The paper addresses a significant challenge in transfer learning: the selection of the optimal pre-trained model for a designated downstream task.
2.The correlation between neural collapse in pre-trained models and their subsequent fine-tuned versions is intriguing and serves as the foundation for the method proposed.
3.The introduced FaCe method is innovative and incorporates both class separation and class fairness, potentially preventing biases during model selection.
4.The array of experiments span multiple tasks and training methodologies, underscoring the robustness and universality of FaCe.

**Weaknesses:**

1.The introduction contains repetitive statements regarding transferability estimation and its objectives. For clarity and brevity, such redundancy should be avoided.
2.The meaning of the variable 't' in Equation (6) is not defined in the surrounding context.
3.When calculating the Variance Collapse, the authors assign equal weight to each category. However, during performance evaluation, categories with larger sample sizes play a more significant role. To ensure FaCe genuinely represents the potential performance of pre-trained models on the target dataset, the current setup seems somewhat flawed. An explanatory note from the authors is sought.
4.In the process of computing and presenting equidistance, the class fairness score F only explores equal distances of each class to the remaining classes. Due to the domain shift, features don't lie on a unit sphere. Thus, "equal distance of each class to the other classes" is not synonymous with "equal distances amongst all classes." The claim "Due to the domain shift, the features do not lie on the unit sphere" lacks sufficient justification. A detailed explanation from the authors would be appreciated.
5.The paper's primary contribution builds upon Variance Collapse by incorporating Class Fairness for a more accurate assessment of model fairness across classes. Yet, the experimental section lacks ablation studies on Class Fairness. To substantiate the effectiveness of the method, the inclusion of relevant experimental evidence is imperative.

**Questions:**

1.While employing FaCe as the criterion to judge the generalizability of pre-trained models in target domains, have the authors considered the impact of differences in class distributions and class counts between source and target domains?
2.The authors extended the concept of equiangularity to equidistance. In the process of calculating distances between classes, why was the current method chosen? Were alternative approaches contemplated?

---

> ### Author Response · Authors · 2023-11-19
>
> We thank the reviewer for providing valuable comments and address your concerns in the following responses.
>
> > **[Q1]**: The introduction contains repetitive statements regarding transferability estimation and its objectives. For clarity and brevity, such redundancy should be avoided.
>
> **[A1]** Thanks for your suggestion, we have updated the related description in the Introduction.
>
> > **[Q2]**: The meaning of the variable 't' in Equation (6) is not defined in the surrounding context.
>
> **[A2]** Sorry for the typos, the variable t is the temperature of softmax, we have updated the related description in Section 3.2.
>
> > **[Q3]**: In Variance Collapse calculation, equal weight is given to all categories, yet performance evaluation gives more importance to larger sample size categories. This setup may affect the representation of FaCe's performance on the target dataset.
>
> **[A3]** The final fine-tuned accuracy is computed on the test set. In practice, we may not know the class distribution of the test set when calculating the score, so it's not possible to pre-assign weights to certain classes. What we can obtain is the class distribution of the training set. In downstream tasks, to ensure generalization on the test set, the class distribution of the training set is often balanced.
>
> > **[Q4]**: "equal distance of each class to the other classes" is not synonymous with "equal distances amongst all classes." The claim "Due to the domain shift, the features do not lie on the unit sphere" lacks sufficient justification.
>
>
> **[A4]** When Neural Collapse (NC) occurs, the features of the training data collapse to class means, and the class means, along with the classifier weights, converge to the vertices of a simplex equiangular tight frame (ETF). At this point, the class means of the features can be considered as distributed on a hypersphere [1]. However, the model is pre-trained on the source data. On downstream target data,  domain shift exists, and the class means will not be located on the hypersphere.
>
>
> Our class fairness term is inspired by NC2. A widely-used metric in existing literatures is defined as $$ \mathcal{NC}_2(W) = \left\|\frac{W W^{\top}}{\left\|{W} {W}^{\top}\right\|_F}-\frac{1}{\sqrt{K-1}}\left({I}_K-\frac{1}{K} \mathbf{1}_K \mathbf{1}_K^{\top}\right)\right\|_F.$$ Essentially, this metric measures the equiangularity of the classifier weights or class means. To use this metric, it should be under the scenarios where the class means are located on the hypersphere.
>
> Hence, we chose not to use the equiangularity measure of NC2 as found in existing work but instead propose to approximate NC2 using the equidistance of each class distribution.
>
> We apologize for any confusion caused by the unclear description.  We have clarified the relevant description in Section 3.2.
>
> - [1] Neural Collapse: A Review on Modelling Principles and Generalization. TMLR 2023.
>
>
> > **[Q5]**: The experimental section lacks ablation studies on Class Fairness. To substantiate the effectiveness of the method, the inclusion of relevant experimental evidence is imperative.
>
> **[A5]** We agree that the ablation study is crucial. As shown in Section 4.6, we provide the ablation study of two terms of FaCe. The results validate that both terms are effective, and they complement each other to achieve the best average performance.
>
> > **[Q6]**: While employing FaCe as the criterion to judge the generalizability of pre-trained models in target domains, have the authors considered the impact of differences in class distributions and class counts between source and target domains?
>
> **[A6]** In transferability estimation, only the pre-trained source model is available, and the information about the source data is unknown. We believe that the only source model available is a more practical setup. In this setting, we cannot obtain the class distribution of the source or consider differences between the source and target. Your idea is reasonable, and we will consider about it in future work.

---

> ### Author Response · Authors · 2023-11-19
>
> > **[Q7]**: The authors extended the concept of equiangularity to equidistance. In the process of calculating distances between classes, why was the current method chosen? Were alternative approaches contemplated?
>
> **[A7]** The most naive approach is to represent a class using its class mean and compute the class mean (CM) distances. However, due to significant differences in the intra-class variances of different classes in the target data, it is more reasonable to fit Gaussian distributions to the feature distributions of different classes and calculate the overlap. When measuring overlap between distributions, the introduction of Bhattacharyya (Bh) distance is a common practice.
>
> We provide experiments using class mean distances as an alternative to measuring class distribution overlap below. Table (a) (b) and \(c\) correspond to Tables 1 2 and 3,  in our paper, respectively (*i.e.*, (a) heterogeneous model zoo with a single source, (b) heterogeneous model zoo with multiple sources, and \(c\) homogeneous model zoo with multiple sources and loss functions). The values in the table are the average results. It can be seen that FaCe with Bhattacharyya distance (FaCe W/ Bh) has a better average performance.
>
>
> | (a) | FaCe w/ Bh | FaCe w/ CM |
> |:---:|:---:|:---:|
> | Kendall | 0.56  | 0.32  |
> | Pearson | 0.62  | 0.37  |
>
> | (b) | FaCe w/ Bh | FaCe w/ CM |
> |---|:---:|:---:|
> | Kendall | 0.62  | 0.55  |
> | Pearson | 0.78  | 0.44  |
>
> |\(c\)| FaCe w/ Bh | FaCe w/ CM |
> |:---:|:---:|:---:|
> | Kendall | 0.38  | 0.36  |
> | Pearson | 0.37  | 0.73  |

---

> ### Author Response · Authors · 2023-11-21
> **Invitation to further discussion**
>
> Dear reviewer,
>
> We genuinely appreciate the time and effort you've invested in reviewing our paper. We have carefully provided relevant responses and results to your concerns. We are eager to further discuss with you and gain your insights **before the end of the Author/Reviewer phase**. Please let us know if any aspect of our work remains unclear or if you have additional feedback.
>
> Thank you.

---

> ### Author Response · Authors · 2023-11-23
>
> Dear Reviewer,
>
> Since the discussion deadline is approaching in less than 11 hours, we kindly request your feedback on whether the response adequately addresses your concerns. If you have any more questions, we would be happy to provide further clarification.
>
> Your timely response is greatly appreciated.
>
> Thanks.

---

### Official Review · Reviewer_cirw · 2023-11-02

**Soundness:** 2 fair
**Presentation:** 2 fair
**Contribution:** 2 fair
**Rating:** 5
**Confidence:** 5

**Summary:**

Prior studies have revealed that well-trained models exhibit the phenomenon of Neural Collapse (NC). The authors observe a strong correlation between the neural collapse of pre-trained models and their corresponding fine-tuned models. Considering the three characteristics of NC, the authors propose the Fair Collapse (FaCe) metric to help select pre-trained models that perform better after fine-tuning. FaCe consists of two key components: variance collapse term and class fairness term. The second components is the key contribution.

**Strengths:**

- The authors explore the impact of Neural Collapse (NC) in the "pre-training then fine-tuning" paradigm and observe that the ranking of NC in the pre-trained models remains mostly consistent during the fine-tuning process.
- The authors employ a metric Fair Collapse (FaCe) to estimate the transferability of pre-trained models.

**Weaknesses:**

- The first term of NC is common. This idea is commonly used in various classification tasks.
- The class fairness score F is used to make any class distribution has a similar overlap with the distribution of other classes. This is quite similar to making the distances between these distributions equal.
- The fine-tuning hyperparameters of different pre-trained models may have a significant impact, and the phenomena observed in the paper might lack persuasiveness.
- The paper lacks experiments on the relationship between the accuracy of pre-trained models, the accuracy of fine-tuned models, and the proposed FaCe method.

Although the authors observed an interesting phenomenon, it may not be solid. They have not clarified the differences between their metric and those presented in other papers. All in all, at this point in time, I would recommend this paper as weak reject.

**Questions:**

see weaknesses

---

> ### Author Response · Authors · 2023-11-19
>
> We thank the reviewer for providing valuable comments and address your concerns in the following responses.
>
> > **[Q1]**: The first term of NC is common. This idea is commonly used in various classification tasks.
>
> **[A1]** We agree that the first term of NC is commonly used in various classification tasks. However, our work introduces the first term of NC as a metric during the downstream fine-tuning process, as shown in Figure 1. Motivated by the observation that the NC of the pre-trained model is correlated with the corresponding fine-tuned model, we include the first term of NC, i.e., the variance collapse term, for transferability estimation, which could be considered as a baseline method of the proposed FaCe.
>
> Even for this term, we want to clarify the difference between the variance collapse term and several closely related works, Hscore [1] considers between-class variance and feature redundancy; GBC [2] is based on the summation of the pairwise class separability. Our variance collapse term is the magnitude of between-class covariance compared to within-class covariance, which is different from them.
>
> - [1] An information theoretic approach to transferability in task transfer learning
> - [2] Transferability Estimation Using Bhattacharyya Class Separability
>
>
> > **[Q2]**: Class fairness score F is quite similar to making the distances between these distributions equal.
>
> **[A2]** The high-level idea that makes the distances between class distributions equal is indeed used by some classification methods. However, such a perspective has not been explored in the transferability estimation task.
> Also, we do not directly borrow ideas from other works but design a specific metric for our task.
>
> The main insight lies in the observation that the NC metrics are robust along the fine-tuning process, motivating us to estimate the degree of NC to estimate the fine-tuned accuracy, i.e., the transferability.
>
> Different from the score in NC2, we propose a similar formulation that estimates equidistance based on the distribution overlap matrix entropy. We also provide a comparison with NC2 in Appendix A.2, the results validate that our method is more suitable for transferability estimation problem and achieve a higher performance.
>
>
>
> > **[Q3]**: The fine-tuning hyperparameters of different pre-trained models may have a significant impact, and the phenomena observed in the paper might lack persuasiveness.
>
> **[A3]** We follow existing works [1,2,3] in transferability estimation and select a set of hyperparameters that perform best on the downstream dataset.  In fact, the results are obtained based on different seeds, indicating the robustness of the fine-tuning process. We have clarified this point in the experimental details of Section 4.1.
> - [1] Transferability Estimation using Bhattacharyya Class Separability.
> - [2] LEEP: A New Measure to Evaluate Transferability of Learned Representations.
> - [3] LogME: Practical Assessment of Pre-trained Models for Transfer Learning.
>
> > **[Q4]**: lacks experiments on the relationship between the accuracy of pre-trained models, the accuracy of fine-tuned models, and the proposed FaCe.
>
> **[A4]** In revision, we updated the zip file of supplementary materials, which added a file named "Acc-FaCe.xlsx." This file provides the accuracy and FaCe values of our experiments. Besides, Figure 4 presents the qualitative results of the fine-tuned model accuracy and FaCe Score, along with comparisons to other methods. However, the classifier dimension of the pre-trained model does not match the number of classes in the downstream target data, making it impossible to provide accuracy directly for the pre-trained model.
>
> > **[Q5]**: The authors observed an interesting phenomenon, it may not be solid.
>
> **[A5]** We agree. Our main insight is based on a preliminary study, motivating us to develop a new score FaCe. The two terms of FaCe correspond to two characteristics of NC. To validate the effectiveness, we conducted numerous experiments, and found it works quite well across various model zoos and tasks. We believe the proposed method FaCe would shed some light on future work.
>
>
> > **[Q6]**: They have not clarified the differences between their metric and those presented in other papers.
>
> **[A6]** We feel sorry for any confusion. As written in Section 2, previous works focused on between-class separability and the information of features, which is similar to the high-level idea of our variance collapse term.
>
> Compared to existing methods, we further consider the class fairness of pre-trained models towards the target class, this is a new perspective, which has not been explored by existing methods.

---

> ### Author Response · Authors · 2023-11-21
> **Invitation to further discussion**
>
> Dear reviewer,
>
> We genuinely appreciate the time and effort you've invested in reviewing our paper. We have carefully provided relevant responses and results to your concerns. We are eager to further discuss with you and gain your insights **before the end of the Author/Reviewer phase**. Please let us know if any aspect of our work remains unclear or if you have additional feedback.
>
> Thank you.

---

> ### Author Response · Authors · 2023-11-23
>
> Dear Reviewer,
>
> Since the discussion deadline is approaching in less than 11 hours, we kindly request your feedback on whether the response adequately addresses your concerns. If you have any more questions, we would be happy to provide further clarification.
>
> Your timely response is greatly appreciated.
>
> Thanks.

---

### Official Review · Reviewer_pEKS · 2023-11-03

**Soundness:** 4 excellent
**Presentation:** 4 excellent
**Contribution:** 3 good
**Rating:** 6
**Confidence:** 2

**Summary:**

Inspired by Neural Collapse, the paper proposes a new metric, FaCe for transferability estimation. In addition to intra-class and iter-class distance, FaCe utilizes the third condition: all classes should be evenly spread in feature spaces. The third condition is evaluated by Bhattacharyya coefficient between class distributions, which is named Class Fairness. FaCe outperforms other methods in diverse benchmarks for transferability estimation.

**Strengths:**

- Motivation is clear and well supported by method design.
- Paper is well organized and easy to understand
- The performance of FaCe looks promising.

**Weaknesses:**

- Between-class covariance, $\Sigma_B$, is defined as the distance between the class avg and the global avg, which does not align with the definition of between-class, a distance between classes.

- Class Fairness term might be related to Variance Collapse term. Class Fairness term uses intra-class and inter-class covariance, which are also used in Variance Collapse term. But, the relationship between the two terms is not studied enough.

**Questions:**

- Why does between-class covariance $\Sigma_B$ in Variance Collapse use global average $h_G$ instead of class average $h_k$? I believe $\Sigma_B=\frac{1}{K} \Sigma_{k_i} \Sigma_{k_j} (h_{k_i} - h_{k_j})^2 $ would be more correct for the between-class covariance than $h_G$.

- Bhattacharyya coefficient looks similar to Variance Collapse term. Bhattacharyya coefficient is a multiplication between inverse within-class covariance and between-class covariance, while Variance Collapse is a multiplication of within-class covariance and inverse between-class covariance. How about replacing Variance Collapse with inverse Bhattacharyya coefficient for simplicity of formulation?

---

> ### Author Response · Authors · 2023-11-19
>
> We thank the reviewer for providing valuable comments and address your concerns in the following responses.
>
> >  **[Q1]**: Between-class covariance, $\Sigma_B$, is defined as the distance between the class avg and the global avg, which does not align with the definition of between-class, a distance between classes.
>
> **[A1]** Here, we directly adopt the common definition of variance. Intuitively, the distance between the class avg and the global avg in $\Sigma_B$ serves the purpose of quantifying the total distance between all class means and the global mean. When the distances between classes are large, the class means will be farther away from the global mean, and when the distances are small, they will be closer to the global mean. Therefore, $\Sigma_B$ can effectively reflect the distance between classes.
>
> > **[Q2]**: Class Fairness term might be related to the Variance Collapse term. The Class Fairness term uses intra-class and inter-class covariance, which are also used in the Variance Collapse term. However, the relationship between the two terms is not studied enough.
>
> **[A2]** Variance Collapse score (VC) is the magnitude of between-class covariance compared to within-class covariance. Different from that, the covariance in Class Fairness score (CF)  is used to estimate the parameters of Gaussian distributions when modeling class features. These two terms complement each other, and we have conducted ablation experiments in Table 4 to investigate their individual effects.
>
> Thanks for your suggestion, we have included some explanations of the relationship between the two terms in Section 4.6.
>
> > **[Q3]**: Why does between-class covariance $\Sigma_B$ in Variance Collapse use global average $h_G$ instead of class average $h_k$? I believe $$ \Sigma_B=\frac{1}{K} \Sigma_{k_i} \Sigma_{k_j} (h_{k_i} - h_{k_j})^2 $$ would be more correct for the between-class covariance than $h_G$.
>
> **[A3]** Thanks for your suggestion. $\Sigma_B$ in Eq. (3) measures the sum of distances between all class means and the global mean, while the $\Sigma_B$ you mentioned measures the sum of distances between pairwise class means. These two forms may be influenced by class imbalance, but they are equivalent when the classes are balanced. We simplify the notation $k_i$ and $k_j$ as $i$ and $j$. The proof is provided below.
> $$
> \begin{align}
> \Sigma_B = &\frac{1}{K} \Sigma_{i} \Sigma_{j} (h_{i} - h_{j})^2 \\
> = & \frac{1}{K} \Sigma_{i} \Sigma_{j} (h_{i} - h_G + h_G- h_{j})^2 \\
> = & \frac{1}{K} \Sigma_{i} \Sigma_{j} [ (h_{i} - h_G)^2 + (h_G- h_{j})^2 + 2(h_{i} - h_G)(h_G- h_{j}) ] \\
> = & \frac{1}{K} \Sigma_{i}K(h_{i} - h_G)^2 + \frac{1}{K}\cdot K\cdot \Sigma_{j} (h_{j} - h_G)^2 + 2 \cdot \frac{1}{K} \Sigma_{i}\Sigma_{j}(h_{i} - h_G)(h_G- h_{j}) .
> \end{align}
> $$
> When the number of each classes is balanced, we have $h_G = \frac{1}{K}\Sigma_{k_j} h_{k_j} = \frac{1}{K}\Sigma_{k_i} h_{k_i}$, The third term above can be transformed into:
> $$
> \begin{align}
> \frac{2}{K} \Sigma_{i}\Sigma_{j}(h_{i} - h_G)(h_G- h_{j})
> & = \frac{2}{K}\cdot\Sigma_{i}(h_{i} - h_G)\Sigma_{j}(h_G- h_{j}) \\
> & = \frac{2}{K}\cdot\Sigma_{i}(h_{i} - \frac{1}{K}\Sigma_{j} h_{j} )\Sigma_{j}(\frac{1}{K}\Sigma_{i} h_{i} - h_{j}) \\
> & = \frac{2}{K}\cdot(\Sigma_{i}h_{i} - K\cdot\frac{1}{K}\Sigma_{j}h_{j} ) \cdot (K\cdot\frac{1}{K}\Sigma_{i}h_{i} - \Sigma_{j}h_{j}) \\
> & = 0
> \end{align}
> $$
> Therefore,
> $$
> \begin{align}
> \Sigma_B = \frac{1}{K} \Sigma_{i}K(h_{i} - h_G)^2 + \frac{1}{K}\cdot K\cdot \Sigma_{j} (h_{j} - h_G)^2 = 2 \Sigma_{i}(h_{i} - h_G)^2  = 2K \cdot \frac{1}{K}\Sigma_{i}(h_{i} - h_G)^2
> \end{align}
> $$
> In our task, fine-tuning accuracy is measured on the test set. To ensure generalization, the training set is typically class-balanced.
>
> We provide the comparisons of these two methods on the heterogeneous model zoo with a single source (Table 1 in our paper) below. Method "FaCe w/ global cov'' is our $\Sigma_B$, and FaCe w/ btw-distance is your proposed one, the results show that there is minimal difference between the two.
>
> |  | Method | CIFAR10 | CIFAR100 | Pets | CUB | gtsrb | DTD | STL10 | Avg. |
> |---|:---:|:---:|:---:|:---:|:---:|:---:|:---:|:---:|:---:|
> | Kendall | FaCe w/ global cov | 0.81  | 0.83  | 0.39  | 0.33  | 0.10  | 0.56  | 0.90  | 0.56  |
> |  Kendall | FaCe w/ btw-distance | 0.83  | 0.83  | 0.39  | 0.33  | 0.10  | 0.57  | 0.90  | 0.57  |
> | pearson | FaCe w/ global cov | 0.89  | 0.85  | 0.64  | 0.39  | -0.05  | 0.71  | 0.91  | 0.62  |
> |  pearson | FaCe w/ btw-distance | 0.89  | 0.85  | 0.63  | 0.39  | -0.06  | 0.71  | 0.91  | 0.62  |

---

> ### Author Response · Authors · 2023-11-19
>
> >  **[Q4]**: How about replacing Variance Collapse with inverse Bhattacharyya coefficient for simplicity of formulation?
>
> **[A4]** The Bhattacharyya coefficient measures the overlap between class distributions and quantifies the separation between them. The intuitive difference between the Bhattacharyya coefficient and Variance Collapse score is that VC additionally considers the compactness within classes.
>
> Your idea is reasonable as well. We provide the comparisons of these two methods on the heterogeneous model zoo with a single source (Table 1 in our paper) below.  FaCe w/ VC is the original version of FaCe, and FaCe w/ neg-Bh is a variant that replaces our variance collapse term with the sum of the negative Bhattacharyya coefficient between classes. Specifically, we replace the score $C$ in FaCe with: $$ C_{negBh}=- \sum_{i=1}^{K}\sum_{j=1，i\neq j}^{K} B(k_i,k_j).$$
> The results are shown below. Our ultimate goal is to consider models that exhibit both large inter-class distances and relatively uniform class distributions as better. While replacing the first term with other forms may have some impact on the final results, the effect is not significant.
>
> |  |  | CIFAR10 | CIFAR100 | Pets | CUB | gtsrb | DTD | stl10 | Avg. |
> |:---:|:---:|:---:|:---:|:---:|:---:|:---:|:---:|:---:|:---:|
> | Kendall| FaCe w/ VC | 0.81  | 0.83  | 0.39  | 0.33  | 0.10  | 0.56  | 0.90  | 0.56  |
> | Kendall | FaCe w/ neg-Bh | 0.81  | 0.89  | 0.41  | 0.23  | -0.05  | 0.61  | 0.83  | 0.53  |
> | Pearson | FaCe w/ VC | 0.89  | 0.85  | 0.64  | 0.39  | -0.05  | 0.71  | 0.91  | 0.62  |
> | Pearson  | FaCe w/ neg-Bh | 0.88  | 0.91  | 0.64  | 0.57  | -0.01  | 0.58  | 0.78  | 0.62  |

---

> > ### Comment · Reviewer_pEKS · 2023-11-20
> >
> > Thank you for your response.
> >
> > I was stupid about the question related to between-class distance.
> > I appreciate your kind correction.
> >
> > All of my concerns have been solved by your response.
> >
> > But, I'm not familiar with "Transferability Estimation". So, I will not strongly defend my rating.
> > I hope you solve other reviewers' concerns.

---

> > > ### Author Response · Authors · 2023-11-20
> > >
> > > We are delighted to know that the concerns are fully addressed. We do appreciate the time you spent on our manuscript.

---

### Author Response · Authors · 2023-11-22
**Eager to discuss with the reviewers before the deadline**

Dear Reviewers,

Thank you once again for your valuable feedback on our paper and significant efforts on ICLR 2024.
We have diligently addressed every concern and question raised by the reviewers in our author response and the updated paper (please download the PDF file again for the revision).

As the discussion deadline is approaching within a day, we kindly request the reviewers to re-evaluate our work considering our latest responses. If there are any additional questions or concerns from the reviewers, we are more than willing to provide further clarification or responses. Meanwhile, if you find that we have adequately addressed your concerns, we kindly request you consider raising the evaluation score for our paper.

Your feedback is crucial to enhancing our work, and we eagerly anticipate your thoughts.

Best Regards,

The Authors

---

### Author Response · Authors · 2023-11-23
**Thanks for the positive recognition from the reviewers**

We would like to express our gratitude to the reviewers for their positive recognition of our method in terms of **task importance**, **motivation**, **novelty**, and **effectiveness**.

- For task importance, **@Reviewer JPai** indicates that **"the paper addresses a significant challenge in transfer learning"**; **@Reviewer Rkcn** thinks our task has **"potential appeal to a broad audience and its broader implications"**; **@Reviewer TZ7d** thinks **"the paper deals with a hard and open problem"**.

- For motivation, i.e., our observation, **@Reviewer JPai** thinks that **"is intriguing and serves as the foundation for the method proposed."**  **@Reviewer pEKS** thinks our **"motivation is clear and well supported by method design"**.

- For novelty, **@Reviewer Rkcn** agrees that **"the closed-form formulation further shows its strength, with implications both in practical and theoretical contexts"**.  **@Reviewer JPai** thinks that **"the introduced FaCe method is innovative"**.

- For effectiveness, **@Reviewer pEKS** thinks **"the performance of FaCe looks promising."**; **@Reviewer TZ7d** thinks our method **"shows some gains vs other transferability metrics"**, and **"the gains seem clear"** in ablation study.

---

### Author Response · Authors · 2023-11-23
**Clarifying concerns from reviewers who did not participate in the discussion**

Since three reviewers **@Reviewer cirw, JPai, and Rkcn** did not participate in the discussion, we want to clarify some important concerns in their questions.


> Q1: The concern about the novelty of two terms of FaCe from **@Reviewer cirw**.

One of our novelty lies in our interesting observation of the changes in the degree of neural collapse during the fine-tuning process. The two terms in FaCe are motivated by this, rather proposed independently.  **@Reviewer JPai** also thinks our observation **"is intriguing and serves as the foundation for the method proposed."**  **@Reviewer pEKS** thinks our **"motivation is clear and well supported by method design"**.

Although some works in other fields may share high-level ideas with us, we do not borrow them in the transferability estimation task directly.

The first term, i.e., the variance collapse term, of FaCe can be seen as a baseline of our method. Even for this term, it is different from existing transferability estimation methods as we illustrated in Section 2 and the response to **@Reviewer cirw**.

The second term, i.e., the class fairness term, is our key contribution. We first consider the model bias towards each target class in the transferability estimation field, and introduce a novel method based on the proposed distribution overlap matrix entropy. **@Reviewer Rkcn** also agrees that **"the closed-form formulation further shows its strength, with implications both in practical and theoretical contexts"**.  **@Reviewer JPai** thinks that **"the introduced FaCe method is innovative"**.

> Q2: The concern about fine-tuning hyperparameters of different pre-trained models from **@Reviewer cirw**.

We clarify this detail in the revision and the response to **@Reviewer cirw**. Note that **@Reviewer TZ7d** has a similar question, and indicating that our response has **"clarifying an important part of the tuning for the task that I confused"**.

> Q3: The concern about the lack of experiments on the relationship between the accuracy of pre-trained models, the accuracy of fine-tuned models, and the proposed FaCe method from **@Review cirw**.

We include more results in the supplementary material. Note that the accuracy of pre-trained models is not available in the transferability estimation tasks. Since the classifier dimension of the pre-trained model does not match the number of classes in the downstream target data.


> Q4: The concern about class weights from **@Reviewer JPai**.

The final fine-tuned accuracy is computed on the test set, and the class distribution of the test set is unknown since the transferability metric is calculated on the training set. This prevents us from using the given different weights for each class.

> Q5: The concern about the unclear description of domain shift and hypersphere from **@Reviewer JPai and Rkcn**.

We include a detailed explanation in the revision and the response to **@Reviewer JPai and Rkcn**. We apologize for any confusion, and we believe that this does not diminish the novelty and effectiveness of our method.

> Q6: The concern about the ablation study from **@Reviewer JPai**.

We agree that the ablation study is crucial. As shown in Section 4.6, we provide the ablation study of two terms of FaCe. **@Review TZ7d** thinks **"the gains seem clear"** in our ablation study.

> Q7: The debate about whether we should conclude which specific model or architecture best transfers according to the proposed metric from **@Reviewer Rkcn**.

Actually, a qualified transferability estimation method should not exhibit bias towards specific network structures or loss functions. For different tasks, the best model, loss function or source datasets vary.

> Q8: This misunderstanding of the significance of Neural Collapse in our method from **@Reviewer Rkcn**.

We do not assume Neural Collapse in fine-tuning the pre-trained model. By contrast, we study the degree of NC of pre-trained models during fine-tuning, through the process and find an interesting observation. Our method is motivated by this observation, to estimate the degree of NC of the pre-trained model, and it is independent of whether training has reached TPT.

---

### Meta-Review · Area_Chair_wtQa · 2023-12-05

**Metareview:**

The paper received five ratings, all around the borderline. In the discussion phase, a positive reviewer lowered their rating because it was unclear how class fairness improved performance, and Reviewer Rkcn emphasized the concern about the paper's handling of neural collapse. In contrast, unfortunately, the reviewers did not find the strengths exciting enough. Hence, the decision was to not recommend the paper for acceptance.

**Justification For Why Not Higher Score:**

not many strengths even from the positive reviewers

**Justification For Why Not Lower Score:**

Several reviewers raised concerns about novelty + some lingering concerns after the rebuttal

---

### Decision · Program_Chairs · 2024-01-16

Reject